# Association between self-reported sleep duration and serum lipid profile in a middle-aged and elderly population in Taiwan: a community-based, cross-sectional study

Pu Lin,[1] Kai-Ting Chang,[1] Yan-An Lin,[1] I-Shiang Tzeng,[2] Hai-Hua Chuang,[3] Jau-Yuan Chen[1,4]

H-HC and J-YC contributed equally.

[1]Department of Family Medicine, Chang Gung Memorial Hospital Linkou Branch, Taoyuan, Taiwan
[2]Department of Research, Taipei Tzu Chi General Hospital, Buddhist Tzu Chi Medical Foundation, Taipei, Taiwan
[3]Department of Family Medicine, Chang Gung Memorial Hospital Taipei Branch, Taipei, Taiwan
[4]Department of Medicine, College of Medicine, Chang Gung University, Taoyuan, Taiwan

**Correspondence to**
Dr Jau-Yuan Chen;
welins@cgmh.org.tw

## ABSTRACT

**Objectives** The association between sleep duration and serum lipid profile in the middle-aged and the elderly is unclear. The aim of this study was to investigate and evaluate the relationships between sleep duration and levels of serum total cholesterol, low-density lipoprotein cholesterol, high-density lipoprotein cholesterol (HDL-C) and triglycerides in these populations.

**Design** Cross-sectional observational study.

**Setting** Community-based investigation in Guishan Township of northern Taiwan.

**Participants** A total of 400 community-dwelling middle-aged and elderly individuals were enrolled. All participants underwent a baseline assessment in 2014, which included anthropometrics, blood samples and self-administered questionnaires. Participants were classified into three groups based on their sleep duration.

**Outcome measures** Multivariate logistic regression was used to obtain ORs and 95% CIs to assess the relationship between sleep duration and lipid profiles.

**Results** Participant mean age was 64.5 years and 35.3% were men. Subjects with longer (>7 hours) and shorter (<6 hours) nightly sleep duration had a higher prevalence of low HDL-C levels (HDL <40 mg/dL) than those with moderate sleep duration (6–7 hours). Multivariate logistic regression revealed that, compared with individuals with sleep duration of 6–7 hours, the ORs of having low HDL-C were 3.68 (95% CI 1.59 to 8.49) greater for individuals with sleep duration of <6 hours and 2.89 (95% CI 1.10 to 7.61) greater for individuals with sleep duration of >7 hours.

**Conclusions** There was a U-shaped relationship between sleep duration and HDL-C levels. Sleep duration >7 hours or <6 hours increased the risk of low serum HDL-C levels.

## INTRODUCTION

As known, most older adults experience lipid profile disorders.[1] Moreover, the different levels in the percentage of lipid disorders may have an association with demographic factors (such as age and sex).[1] A study also found a significant association between the high prevalence of dyslipidaemia and risk factors, such as increasing age, smoking status, hypertension, diabetes and body mass index.[2] Several studies have identified that longer or shorter sleep duration may increase the mortality and morbidity risks from diabetes mellitus,[3–5] obesity,[6] hypertension[7] and coronary heart disease.[8] Some studies have shown a U-shaped association between sleep duration and major morbidities.[5 9 10] Cardiovascular disease has been a leading cause of death worldwide.[11] Dyslipidaemia, such as high levels of total cholesterol (TC), low-density lipoprotein cholesterol (LDL-C), triglycerides (TG) and low levels of high-density lipoprotein cholesterol (HDL-C), increases the risk of cardiovascular disease,[12] and hence remains an important issue in the field of health promotion and disease prevention. Previous studies have shown age-associated alterations in the level, composition and function of lipid profiles, including an inverted U-shaped quadratic trajectory for TC, LDL-C and TG,[13 14] and a decrease in antioxidative ability.[15] Other than age, factors that have been suggested to be related to lipid profile levels include body mass index (BMI), body composition, diet and cardiorespiratory fitness.[13 14]

### Strengths and limitations of this study

► This is the first study to explore the associations between sleep duration and lipid profiles in middle-aged and elderly Taiwanese population.
► We could not determine a causal relationship in our findings because this was a cross-sectional study.
► Lifestyle questionnaires were self-reported, which may limit the accuracy of measurements.

Although some studies suggest that sleep deprivation may result in changes to plasma lipid levels,[16] there is a lack of consensus on the possible associations that may exist between sleep duration and serum lipid profiles, and even fewer data are available in the Taiwanese population.[17] Therefore, we investigated the relationship between sleep duration and levels of serum TC, LDL-C, HDL-C and TG in a community-based study in Taiwan.

## MATERIALS AND METHODS

### Study participants

The present study was an observational, cross-sectional study. We enrolled 400 participants, including 141 men and 259 women. The inclusion criteria included (1) residents aged >50 years and (2) the residents living in Guishan Township. Twelve people aged <50 years old were excluded from the study. Subjects were excluded if (1) they could not complete the full examinations or had missing data for age, sex, anthropometric values and blood test results; (2) were functionally dependent; (3) were unable to adequately communicate with the interviewers; (4) had major illnesses recently or (5) had known sleep disorders.

### Data collection

Data collection comprised two parts: a physical status examination and a self-administered questionnaire. For the physical status examination, height, weight, abdominal circumference, blood pressure and blood samples were collected for all participants. Blood samples included lipid profiles (TC, LDL-C, HDL-C and TG), aspartate aminotransferase (AST) levels, fasting plasma glucose (FPG) levels, creatinine levels and uric acid levels. The self-administered questionnaire included items related to smoking, drinking, exercise, sleep and other lifestyle habits. Details from all the participants were obtained on the same day while the data collection was performed from March to August 2014.

### Definitions and variables

Based on the National Cholesterol Education Program (NCEP) Adult Treatment Panel (ATP) III guidelines, we set different cut-off points for each item of the serum lipid profiles (LDL-C: 130 mg/dL; TC: 200 mg/dL; HDL-C: 40 mg/dL and TG: 150 mg/dL). Hyperlipidaemia was defined as having LDL-C, TC or TG levels above the cut-off points.

**Table 1** Characteristics of the study subjects with and without hyperlipidaemia

| Variables | No hyperlipidaemia (n=140) | Hyperlipidaemia (n=260) | p Value |
|---|---|---|---|
| Sex (man) | 57 (40.7) | 84 (32.3) | 0.101 |
| Age (years) | 65.4±8.4 | 64.0±8.4 | 0.072 |
| Current smoking | 9 (6.4) | 34 (13.1) | 0.043 |
| Alcohol drinking | 20 (14.3) | 55 (21.2) | 0.107 |
| Regular exercise | 116 (83.2) | 211 (81.2) | 0.681 |
| Hypertension | 64 (45.7) | 137 (52.7) | 0.209 |
| Diabetes | 28 (19.7) | 56 (21.5) | 0.238 |
| LDL-C (mg/dL) | 97.6±17.7 | 129.6±32.6 | <0.001 |
| TC (mg/dL) | 169.8±21.1 | 211.9±33.2 | <0.001 |
| HDL-C (mg/dL) | 55.4±12.8 | 53.9±14.5 | 0.161 |
| TG (mg/dL) | 84.5±29.3 | 142.3±71.6 | <0.001 |
| Sleep (hours) | 6.2±1.2 | 6.2±1.3 | 0.276 |
| SBP (mm Hg) | 128.0±16.0 | 130.2±17.0 | 0.267 |
| DBP (mm Hg) | 76.1±11.3 | 77.1±12.2 | 0.592 |
| BMI (kg/m$^2$) | 23.7±3.4 | 25.0±3.6 | 0.001 |
| WC (cm) | 82.9±8.5 | 86.3±10.1 | 0.001 |
| WHtR | 0.52±0.05 | 0.54±0.06 | 0.003 |
| ALT (mg/dL) | 20.3±8.9 | 23.9±14.6 | 0.032 |
| Creatinine (mg/dL) | 0.8±0.6 | 0.8±0.3 | 0.408 |
| FPG (mg/dL) | 92.2±19.1 | 98.4±28.5 | 0.028 |
| Uric acid (mg/dL) | 5.4±1.3 | 5.9±1.4 | 0.001 |

Data expressed as mean±SD for continuous variables and n (%) for categorical variables.

ALT, alanine transaminase; BMI, body mass index; DBP, diastolic blood pressure; FPG, fasting plasma glucose; HDL-C, high-density lipoprotein; LDL-C, low-density lipoprotein; SBP, systolic blood pressure; Sleep, sleep duration; TC, total cholesterol; TG, triglyceride; WC, waist circumference; WHtR, waist-to-height ratio.

**Table 2** Characteristics of the study subjects by sleep duration groups

| Variable | <6 hours (n=100) | 6–7 hours (n=255) | >7 hours (n=45) | p Value |
|---|---|---|---|---|
| Sex (man) | 30 (30) | 88 (34.5) | 23 (51.1)*,** | 0.044 |
| Age (years) | 65.2±7.6 | 64.1±8.3 | 65.1±10.6 | 0.530 |
| LDL-C (mg/dL) | 115.8±30.4 | 119.2±31.3 | 119.6±40.0 | 0.643 |
| TC (mg/dL) | 197.2±34.9 | 197.0±34.2 | 198.0±45.4 | 0.984 |
| HDL-C (mg/dL) | 57.4±15.8 | 53.9±12.5 | 51.2±16.4 | 0.025 |
| TG (mg/dL) | 120.4±61.8 | 119.8±62.7 | 138.5±88.6 | 0.207 |
| High LDL-C | 33 (33) | 87 (34.1) | 15 (33.3) | 0.978 |
| High TC | 49 (49) | 125 (49.0) | 21 (47.7) | 0.941 |
| Low HDL-C | 13 (13) | 25 (9.8) | 14 (31.1)*,** | 0.001 |
| High TG | 22 (22) | 58 (22.7) | 14 (31.1) | 0.437 |
| Alcohol drinking | 15 (15) | 52 (20.4) | 11 (24.4) | 0.346 |
| Current Smoking | 10 (10) | 27 (10.6) | 6 (13.3) | 0.828 |
| Regular exercise | 80 (80) | 209 (81.9) | 39 (86.4) | 0.642 |
| Hypertension | 47 (47) | 125 (49.0) | 29 (64.4) | 0.122 |
| Diabetes | 25 (25) | 45 (17.6) | 9 (20.0) | 0.293 |
| Hyperlipidaemia | 69 (69) | 161 (63.1) | 30 (66.7) | 0.564 |
| SBP (mm Hg) | 129.9±16.5 | 128.6±16.7 | 133.3±25.2 | 0.216 |
| DBP (mm Hg) | 76.4±13.0 | 76.6±11.8 | 78.9±9.13 | 0.436 |
| BMI (kg/m$^2$) | 24.3±3.9 | 24.6±3.4 | 25.1±3.6 | 0.394 |
| WC (cm) | 84.7±10.4 | 84.8±9.6 | 87.4±8.6 | 0.235 |
| WHtR | 0.5±0.1 | 0.6±0.1 | 0.6±0.1 | 0.417 |
| ALT (mg/dL) | 22.5±14.0 | 22.3±11.94 | 22.7±15.9 | 0.530 |
| Creatinine (mg/dL) | 0.8±0.3 | 0.8±0.3 | 1.0±0.9 | 0.920 |
| FPG (mg/dL) | 98.2±28.2 | 94.8±19.3 | 100.3±45.1 | 0.519 |
| Uric acid (mg/dL) | 5.6±1.3 | 5.8±1.4 | 6.1±1.4 | 0.101 |

Data expressed as mean±SD for continuous variables and n (%) for categorical variables.
*p Value<0.05 compared with <6 hours group, **p Value<0.05 compared with 6–7 hours group.
ALT, alanine transaminase; BMI, body mass index; DBP, diastolic blood pressure; FPG, fasting plasma glucose; high LDL-C, low-density lipoprotein ≥130 (mg/day); high TC, total cholesterol ≥200 (mg/day); high TG, triglyceride ≥150 (mg/day); low HDL-C, high-density lipoprotein ≤40 (mg/day); SBP, systolic blood pressure; WC, waist circumference; WHtR, waist-to-height ratio.

Adapted from the Pittsburgh Sleep Quality Index (PSQI),[17] the questionnaire about sleep included the question: 'What was your daily average sleep duration during the past month (not counting the times lying on the bed without sleeping)?'. We divided sleep duration into three groups: <6 hours, ≥6 hours but ≤7 hours (ie, 6–7 hours) and >7 hours. Hypertension was defined as a self-reported hypertension or the use of antihypertensive drugs. Diabetes was defined as a self-reported diabetes mellitus or the use of oral antidiabetic drugs or insulin. Alcohol drinking was defined as drinking alcohol on ≥2 days per week. Regular exercise was defined as exercising ≥2 days per week. Current smoking was defined as the literal meaning.

The institutional review board approved the study and all patients provided written informed consent.

### Statistical analysis

Adequate blood samples and complete questionnaires were collected from all participants. After data were collected, statistical analysis was conducted using the SPSS V.21.0 statistical package (SPSS, Chicago, Illinois, USA). A p Value <0.05 was considered statistically significant. Our statistical analysis included multiple steps. First, characteristics were compared across hyperlipidaemia and sleep duration sub-groups using t-tests, one-way ANOVA and $\chi^2$ tests. Homogeneity examination was performed with ANOVA, and heterogeneous variables were analysed with the Brown-Forsythe test. Second, we measured correlations between serum lipid profiles and related risk factors. Third, logistic regression analyses were conducted to assess the relationship between sleep duration and the risk of high LDL-C, high TC, low HDL-C and high TG levels; additional regression models were adjusted for relevant covariates.

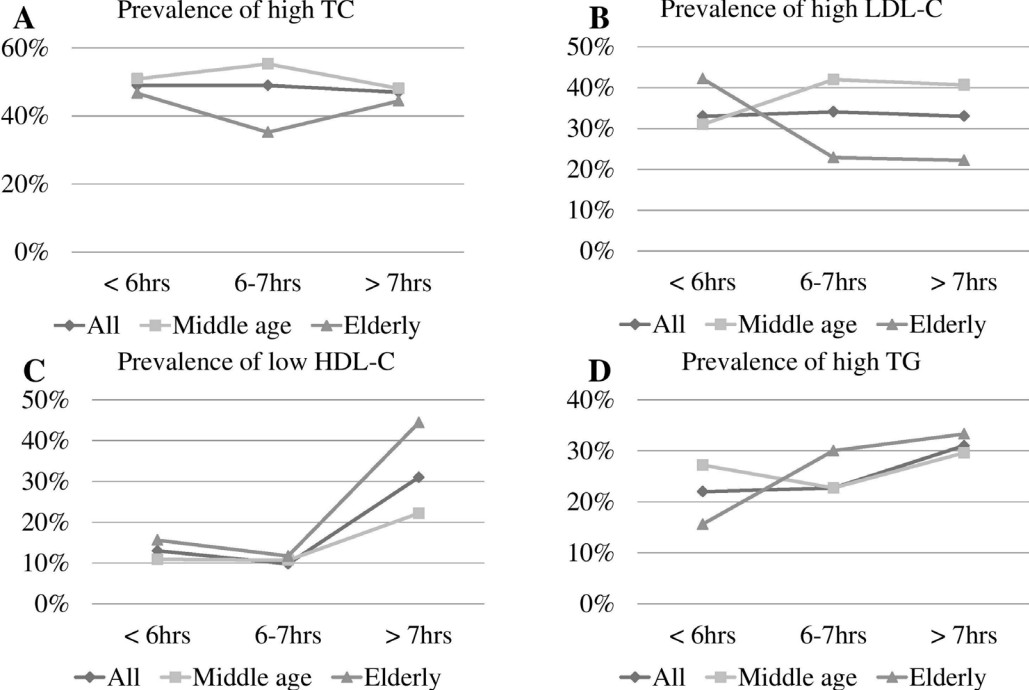

**Figure 1** Prevalence of serum lipid profiles by sleep duration between two age groups (middle age: 50–65 years old; elderly: ≥65 years old). HDL-C, high-density lipoprotein cholesterol; LDL-C, low-density lipoprotein cholesterol; TC, total cholesterol cholesterol; TG, triglyceride.

## RESULTS

The characteristics of study subjects with and without hyperlipidaemia are shown in table 1. Among the included participants, 260 (65%) had hyperlipidaemia. Body mass index, waist circumference, waist-to-height ratio, AST levels, FPG levels and uric acid levels were significantly higher in the hyperlipidaemia group.

The characteristics of the study subjects categorised by sleep duration are shown in table 2. There were 100, 255 and 45 participants in the <6 hours, 6–7 hours and the >7 hours groups, respectively. Low HDL-C levels were significantly higher in the >7 hours group (31.1%) than in the other two groups (9.8% for the 6–7 hours group and 13.0% for the <6 hours group). The post-hoc tests also showed that the difference was significant (significant difference between <6 hours and >7 hours groups (p value=0.009), 6–7 hours and >7 hours (p value=0.001).

Figure 1 shows the comparisons of serum lipid profiles by sleep duration for two age groups: ≥65 years (elderly group, 168 people) and all others (middle age group, 232 people). Abnormal TC prevalence levels were the highest (around 50%) among the four items with no significant differences by sleep duration (figure 1A). The average prevalence of high LDL-C was about one-third, and the prevalence gradually increased with longer sleep duration in the middle age group but decreased with longer sleep duration in the elderly group (figure 1B). There was a U-shaped (or J-shaped) distribution in the association of low HDL-C levels with sleep duration with both age groups showing significantly higher prevalence of low HDL-C when the sleep duration was <6 hours or

>7 hours (figure 1C). Abnormal TG levels by sleep duration in the middle age group also appeared to have a U-shaped distribution (figure 1D).

Multivariable logistic regression model results are shown in table 3. Model 1 assessed the crude OR of lipid profiles by sleep duration; model 2 adjusted for age; model 3 adjusted for age and waist circumference and model 4 adjusted for age, waist circumference and other traditional factors that influence lipid levels, including sex, alcohol drinking, exercise and smoking. The 6–7 hour sleep duration group was set as the reference category. In a model where high TC levels, high LDL-C levels or high TG levels were the outcome, no significant association was found with sleep duration. However, a significant association was found in models 1–4 when the dependent variable was low HDL-C levels (table 3).

## DISCUSSION

In our study, traditional risk factors, such as high BMI, high waist circumference, high FPG levels and high uric acid levels, were significantly associated with abnormal lipid values. Elevated AST levels may be associated with non-alcoholic liver disease. We found a U-shaped association between sleep duration and low HDL-C levels, and logistic regression models showed that sleep duration was significantly associated with low HDL-C levels regardless of the adjustment for potential confounders.

Several studies have reported the associations between serum lipid profiles and sleep duration, but the results of these studies have been inconsistent.[18–23] Choi et al[18]

**Table 3** ORs for the associate between lipid profiles and reported sleep duration

|  | Sleep <6 hours (n=100) | Sleep 6–7 hours (n=255) | Sleep >7 hours (n=45) |
|---|---|---|---|
| **TC** | | | |
| Model 1 | 1.02 (0.54 to 1.92) | 1.00 | 1.10 (0.54 to 2.22) |
| Model 2 | 0.99 (0.52 to 1.88) | 1.00 | 1.10 (0.52 to 2.24) |
| Model 3 | 1.01 (0.53 to 1.93) | 1.00 | 1.13 (0.56 to 2.31) |
| Model 4 | 0.90 (0.45 to 1.80) | 1.00 | 0.95 (0.44 to 2.02) |
| **LDL-C** | | | |
| Model 1 | 1.04 (0.53 to 2.03) | 1.00 | 0.99 (0.47 to 2.08) |
| Model 2 | 1.02 (0.52 to 2.00) | 1.00 | 0.99 (0.47 to 2.10) |
| Model 3 | 1.05 (0.53 to 2.07) | 1.00 | 1.03 (0.49 to 2.19) |
| Model 4 | 1.001 (0.50 to 2.01) | 1.00 | 0.94 (0.43 to 2.04) |
| **HDL-C** | | | |
| Model 1 | 3.81 (1.80 to 8.05)* | 1.00 | 3.02 (1.28 to 7.14)* |
| Model 2 | 3.75 (1.77 to 7.93)* | 1.00 | 3.02 (1.28 to 7.16)* |
| Model 3 | 3.62 (1.63 to 8.00)* | 1.00 | 2.93 (1.17 to 7.34)* |
| Model 4 | 3.68 (1.59 to 8.49)* | 1.00 | 2.89 (1.10 to 7.61)* |
| **TG** | | | |
| Model 1 | 0.65 (0.33 to 1.31) | 1.00 | 0.63 (0.28 to 1.37) |
| Model 2 | 0.65 (0.33 to 1.31) | 1.00 | 0.63 (0.27 to 1.38) |
| Model 3 | 0.72 (0.35 to 1.46) | 1.00 | 0.69 (0.31 to 1.35) |
| Model 4 | 0.64 (0.31 to 1.34) | 1.00 | 0.60 (0.26 to 1.39) |

Data are expressed as OR and 95% CI.

*p Value <0.05.

†Model 1: unadjusted; model 2: multiple logistic regression adjusted for age; model 3: multiple logistic regression adjusted for age and waist circumference; model 4: multiple logistic regression adjusted for age, sex, waist circumference, alcohol drinking, exercise and smoking.

HDL-C, high-density lipoprotein cholesterol; LDL-C, low-density lipoprotein cholesterol; TC, total cholesterol cholesterol; TG, triglyceride.

collected data from 4222 Korean participants over the age of 60 years and found a U-shaped association between low HDL-C levels and high TG levels, which were similar to the results of our study. Both short and long sleep durations were related to an increased risk of metabolic syndrome and sleep duration of 7 hours demonstrated the lowest prevalence of metabolic syndrome in this study. Hall et al[19] reported that sleep duration was independently associated with three components of metabolic syndrome: abdominal obesity, elevated serum glucose levels and elevated TG levels. The optimal sleep duration for preventing metabolic syndrome determined by this study was 7–8 hours per night. Bjorvatn et al[20] demonstrated that cholesterol levels, TG levels and blood pressure were higher in subjects with short sleep duration. Another study[21] revealed that HDL-C levels decreased with short and long sleep duration among normotensive, but not hypertensive, women.

The logistic regression models, adjusted for age, sex, waist circumference, alcohol drinking, exercise and smoking, showed a significant association with low HDL-C levels. To our knowledge, the prevalence of dyslipidaemia is proportional to age and different in men and women. The association between waist circumference and dyslipidaemia has been confirmed in some studies.[24 25] Lifestyle factors have been shown to influence serum lipid or lipoprotein levels. For example, smoking decreases HDL-C levels and increases TG levels, whereas alcohol consumption increases the levels of both.[26–28] Exercise increases HDL-C levels and decreases TG levels.[29 30] In addition, alcohol consumption is reported to decrease LDL-C levels.[31 32] Even after adjusting for these potential confounding factors, our logistic regression models showed significant differences in ORs of low HDL-C levels between sleep duration groups.

The influence that shorter sleep duration has on body weight and dyslipidaemia has become clearer in recent years. Sleep restriction is associated with hormone imbalance; it reduces leptin (an appetite suppressant) and elevates ghrelin levels (an appetite stimulant),[6 33] which may contribute to increased body weight and lead to dyslipidaemia. The biochemical mechanism for the relationship between longer sleep and dyslipidaemia has not been clearly confirmed; some studies showed that prolonged sleep duration may be associated with glucose intolerance and diabetes.[3 34] Physical performance, such as reduced energy consumption due to increased time in bed, may affect obesity; one study explained this relationship by showing that longer sleep duration was related to less exercise.[8]

Some studies have reported that the relative risk of mortality and morbidity was the lowest when sleep duration was 7–8 hours per night,[5 8–10] but other research has reported that 6–7 hours is more optimal.[22] The cut-off point of our study was set at 6–7 hours; we found the volunteers in our study generally woke up early regardless of the amount of time they slept, and they often had spontaneous arousal from nocturnal sleep. This may be explained by physiological age-related changes in circadian modulation, homeostatic factors, cardiopulmonary function and endocrine function[35 36] or it might be due to underlying chronic diseases. In an attempt to recommend the optimal sleep duration for lowering morbidity based on epidemiological data, it is inferred that the optimal sleep duration will vary for different populations.[22]

Limitations of this study should be mentioned. First, since this was a cross-sectional study, a cohort study is required for determining more causal relationships. Second, selection bias might exist because the volunteer participant selection was limited to one township, and participants were not selected randomly or with a stratified method from the population. This could limit the generalisability of these results. In addition, our sample size was relatively small; hence, sampling bias should be considered. Third, we did not group the population by underlying diseases, such as diabetes or cerebrovascular disease, due to the limited sample size. The proper cut-off for lipid profiles may differ according to underlying

diseases. Furthermore, the HDL-C cut-off point differs by sex; in metabolic syndrome, it is set as 40 mg/dL for men and 50 mg/dL for women. The influence of sleep duration on lipid metabolism may also differ between men and women; in a study by Kaneita *et al*,[22] among 1666 men and 2329 women aged ≥20 years from Japan, it was shown that HDL-C levels had a U-shaped association with sleep duration in women but not in men. Fourth, glycosylated haemoglobin might be a more appropriate measurement of diabetes instead of self-reported diabetes mellitus or the use of oral antidiabetic drugs or insulin. Fifth, similar to the majority of studies in the literature examining the effects of sleep duration on serum lipids, diet was not factored in as a covariate. It may be explained that daily life diet is a diversified condition. Thus, it may be difficult to represent it with just one variable in order to adjust for it in the regression model. Sixth, lipid-lowering therapy was an important factor influencing lipid levels, but we did not consider lipid-lowering medications while formulating the questionnaire. Finally, the present study estimated sleep duration only by a self-reported questionnaire, which is a more systematically biased estimate than measured sleep duration.

## CONCLUSIONS

In our study of a Taiwanese population, sleep duration of 6–7 hours per night reduced the risk of abnormal serum lipid profiles. Sleep duration over 7 hours or <6 hours may increase the risk of low serum HDL-C levels, although adjustment for risk factors attenuated this relationship. Inappropriate sleep duration might be a potential risk factor for low HDL-C levels, and adequate sleep duration may improve low HDL-C status. Lifestyle interventions, including exercise and abstaining from alcohol and smoking, should be initiated early in high-risk groups.

**Contributors** PL: involved in the data collection, analysis and writing of the manuscript. KTC and YAL: involved in the collection of data. IST: provided statistical advice. HHC and JYC: contributed to the conceptualisation; they designed, performed the experiments, collected and analysed the data, as well as revised it critically for important intellectual content; and also approved the final version to be submitted.

**Funding** This work was supported by Chang Gung Memorial Hospital (CORPG3C0011, CORPG3C0171, CORPG3C0172, CZRPG3C0053).

**Competing interests** None declared.

**Ethics approval** The study was approved by Chang-Gung Medical Foundation Institutional Review Board (102-2304B), and written informed consent was given by all the participants before enrollment.

**Provenance and peer review** Not commissioned; externally peer reviewed.

**Data sharing statement** No additional data are available.

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
