## [Reviewer comments · BMJ Open]

ARTICLE DETAILS

TITLE (PROVISIONAL)	Association between self-reported sleep duration and serum lipid profile in a middle-aged and elderly population in Taiwan: A community-based, cross-sectional study
AUTHORS	Lin, Pu; Chang, Kai-Ting; Lin, Yan-An; Tzeng, I-Shiang; Chuang, Hai-Hua; Chen, Jau-Yuan

VERSION 1 – REVIEW

REVIEWER	Felipe Beijamini Federal University of Fronteira Sul - Realeza/PR - Brazil
REVIEW RETURNED	06-Apr-2017

GENERAL COMMENTS	This is an interesting manuscript aiming to evaluate the association between serum lipid profile and sleep duration in an elderly population. There is a lack of studies testing such association specifically in elderly people. The main findings are a U-shaped association between sleep duration and low-levels of HDL cholesterol. Although the result is interesting I have specific concerns regarding the methods and statistical analysis performed which are detailed described bellow. Abstract: The abstract needs clarification. The section about participants must state a description of the sample and not about the methods that have been used. Authors should be careful about the sentence "participants were divided into.." no one was divided, but classified according to the categories for sleep duration and lipid profile. Introduction The introduction is a bit too short. I would like to see more information about the lipid profile in elderly population. What are the main factors that can change it? How's the prevalence of hyperlipedemia in elderly population? Furthermore, authors could develop a better dialogue of the literature discussing about the sleep duration of this population. There are several reports about sleep duration in elderly population from Taiwan. Authors should present this data to prepare the reader for their data and for the comparison on discussion section. Methods Sample: there is no clear information about the recruitment. Also, I could not find detailed information about inclusion/exclusion criteria. Authors must clarify this. It is not clear if this is a study including or not people presenting any cardiometabolic disease, for example. This is one of the factors that might be driven the main findings of the manuscript.
---

Materials and variables: the question about sleep duration is very like the question about sleep duration from the Pittsburgh Sleep Quality Index (PSQI). If the PSQI was applied authors must mention it. If not, author must explain why are they interested in the mean of sleep duration in the past month.

Regarding the categorization of sleep duration there is no explanation why three groups and no explanation about the criteria for the establishment of three categories. Why not 5 categories? Why not different sleep durations as grouping factors? Authors could use the NSF recommendation for sleep duration as criteria, however, it would be even better if they presented the distribution of sleep duration in this population and based their categorization according to the frequency distribution in this population.

The categorization for serum lipid profiles also could be better explained. Considering the NCEP ATP III guidelines there are several categories for each lipid (low, normal, borderline, high or very high). Authors decided by a binary categorization having just hyperlipidemia or no hyperlipidemia as categories. I'm a bit concern about this categorization and the lack of information in this case.

Authors should take advantage of the sample-size and explore in more detail the lipid profile of their population. Furthermore, authors could perform analysis considering the data as a continuous variable and not as a grouping factor.

Regarding the questionnaires, how is it? How was evaluated the alcohol intake? Same for smoking and exercise. It is not clear how these factors were accessed. Furthermore, was there any evaluation of cardiometabolic disturbances? If so, authors must make it clear at manuscript.

Procedures: there is no clear description about the timeline of the study. Were the evaluations (physical, blood sampling, and questionnaires) performed at same day for every subject? If not, please describe it.

Statistical analysis and results:

The main statistical analysis are ok for the aim of the study. However, I have a few concerns regarding the strategy authors have use Spearman correlation to evaluate association between the serum lipid profile and several variables such as sex, current smoking and exercise. These variables are (or seem to be) binary. How could Spearman's correlation being performed for it? Still in regard of the correlation analysis: if the association between sleep duration and HDL is a U-shaped curve, how can authors explain the negative correlation presented at Table 3?

Regarding the Figure 1 the X axis labels are different from the statistical analysis. Please verify.

I will make no further comments on Discussion and conclusion considering my concerns on methods.

REVIEWER	Safwaan Adam 1) University of Manchester 2) Central Manchester University Hospitals NHS Trust 3) Salford Royal NHS Foundation Trust All 3 based in United Kingdom
REVIEW RETURNED	15-May-2017

GENERAL COMMENTS	Well conducted study with nice simple study design and clear objective. Evidence regarding the primary aim from this geographical region is limited. The association between sleep duration and cardiometabolic illness itself however is described in the literature with some gaps in the knowledge base remaining. This study will contribute to the current evidence with the added advantage of being in a community setting. There are some minor but important changes (mainly clarifications) needed and I have detailed these. Line 7: Reference '10' has been cited for this sentence. This citation seems more suited to the next sentence and should be placed on line 9. Line 13 and 14: '400' participants were enrolled however the numbers in the next line add up to 391 participants (141 men and 250 women). Please account for the apparent discrepancy. Were there any specific inclusion/exclusion criteria for participants? Were patients with known sleep related disorders also included (for example obstructive sleep apnoea). Van den Berg et al. (reference 16) in the discussion section of their paper did suggest that sleep apnoea did influence their results (albeit in a minor way) however there is ample literature available detailing the effects of sleep apnoea on cardiometabolic health. It is important to denote if patients with this condition were included or not. Line 18 and 19: Blood sampling was done to measure for lipid profile, AST levels, fasting plasma glucose, creatinine and uric acid levels. In this population, using the numbers given in the table, the prevalence of diabetes was 20%. Glycated haemoglobin would have been a more appropriate measurement of glycaemia especially given the already high prevalence of diagnosed diabetes in the group. Are HbA1c results available to report? Line 21: 'Other lifestyle habits': Were there any questions relating to dietary habits? I notice in this study as well as other similar studies in the literature covering examining effects of sleep duration on serum lipids, diet does not seem to be factored for as a covariate. Whereas I appreciate the difficulty in trying to ascertain dietary habits in a study such as this one, was there any information at all available? Perhaps this could be acknowledged and expanded upon within the discussion. Line 26: The definition of hyperlipidaemia is acceptable but in terms of lipid lowering therapy, what adjustment was made for these during analysis of the data? This is very pertinent given that 65% of participants in this study had a diagnosis of dyslipidaemia so I would assume a fair proportion of the participants were on lipid-lowering therapy. Commonly used drugs such as statins and fibrates can both influence HDL-C which was the main finding reported but additionally use of these drugs would have influenced the results on other lipid profile parameters.
---

The use of lipid lowering therapy should be accounted for when discriminating findings between groups and adjusted for in the regression model.

Line 34:

If possible, could units of alcohol consumed per week be reported as opposed to simply days per week. Though drinking 2 or more days per week is likely to be a very inclusive parameter, there could be binge drinkers that are omitted by excluding units. If alcohol consumption (by units) information is not available, the current definition will suffice.

Line 35:

Likewise, with exercise, a more precise definition would be getting an estimate of hours per week.

Improved precision of the definitions may improve the accuracy of the regression models.

Table 1:

The waist to hip ratio reported between the dyslipidaemic and non-dyslipidaemic group are exactly the same however the p-value is 0.003. Please double check this (whereas I do appreciate that you may have reported it this way to reduce the decimal points, the result does seem odd in its current form).

Line 66:

Please give details of the chosen post-hoc tests (e.g. Bonferroni etc.) in the methods section. In addition, for this point only, please specify where the differences were significant in post-hoc testing, that is, between which groups.

Line 75:

Please include number of participants in each group in this section, i.e., how many were older than 65?

Line 85 and table 3:

Which sex?

Table 3:

It is interesting that the factors that affected HDL-C (apart from sleep duration) also affected triglycerides in an inverse manner. Sleep duration oddly had a negative correlation coefficient for both serum HDL-c and triglycerides (though non-statistically significant) which is not in keeping with the direction of association for the other covariates.

Similarly, in the regression models, though the model for serum triglycerides had a non-significant p-value, the trend was for a reduced risk of high triglycerides in a U-shaped manner whereas for a low-HDL-c, this U-shaped distribution represented an increased risk. This is not in keeping with the known relationship between HDL-C and triglycerides which is inverse and the expected result would have been a similar distribution in this U-shaped curve for a given factor (as in table 3 for variables other than sleep duration). Interestingly, as the authors later cite in the discussion, a study by Choi et al. (reference 11) also alludes to this unexpected relationship when sleep duration is examined as a variable (especially in the over 60 age group). These findings are not supported by the study by Bjorvatn (reference 13) in which the U-shape curves follow a similar direction (for incidence of high triglycerides and low HDL-c according to sleep duration) albeit with less pronounced odds ratios with longer duration of sleep for triglycerides. Whereas in both this study and the study by Choi et al., the difference in triglycerides between groups is not statistically significant, there is a clear tendency for this association and I would welcome a slightly more expanded discussion around this.

	Line 134: One of the references chosen (reference 18) refers to a study done in a paediatric population. Whereas it does support the author statement, a more suitable reference would be from adult medicine for which there are many. Line 137: Please cite references for statements relating to the effects of smoking, alcohol and exercise on the lipid profile. References: Reference 5 – ‘the’ should read ‘The’. Reference 11 – Please d
--	--

VERSION 1 – AUTHOR RESPONSE

Reviewer 1

Comment#1

Abstract:

The abstract needs clarification. The section about participants must state a description of the sample and not about the methods that have been used. Authors should be careful about the sentence “participants were divided into.” no one was divided, but classified according to the categories for sleep duration and lipid profile.

Response#1:

Thanks for the correction about our carelessness. “Divided” changed to “classified”.

Comment#2

Introduction

The introduction is a bit too short. I would like to see more information about the lipid profile in elderly population. What are the main factors that can change it? How’s the prevalence of hyperlipidemia in elderly population? Furthermore, authors could develop a better dialogue of the literature discussing about the sleep duration of this population. There are several reports about sleep duration in elderly population from Taiwan. Authors should present this data to prepare the reader for their data and for the comparison on discussion section.

Response#2:

We thank the reviewer for the comments and the reviewer’s concern is well taken. The changes are listed in Page 4-5 Lines 1-23.

Comment#3

Methods

Sample: there is no clear information about the recruitment. Also, I could not find detailed information about inclusion/exclusion criteria. Authors must clarify this. It is not clear if this is a study including or not people presenting any cardiometabolic disease, for example. This is one of the factors that might be driven the main findings of the manuscript.

Response#3:

Thanks for reminding us about no clearly statement about inclusion/exclusion criteria. The inclusion criteria included (1) the residents aged over 50 years old; (2) the residents living in Guishan township. 12 people aged younger than 50 years old were excluded from the study.

Subjects were excluded if (1) the residents could not complete the full examinations or missing data for age, sex, anthropometric values and blood test results; (2) the residents were functionally dependent; (3) the residents were unable to adequately communicate with the interviewers; (4) the residents had major illness recently; (5) the residents with known sleep disorders.

We have added a paragraph to the "MATERIALS AND METHODS" section in in Page 5 line 25-33.

Comment#4

Materials and variables: the question about sleep duration is very like the question about sleep duration from the Pittsburgh Sleep Quality Index (PSQI). If the PSQI was applied authors must mention it. If not, author must explain why are they interested in the mean of sleep duration in the past month.

Response#4:

We thank for the reviewer's comments and suggestions. As you said, the question is applied by PSQI. We add it in line 35. (Page 6 line 48)

Comment#5

Regarding the categorization of sleep duration there is no explanation why three groups and no explanation about the criteria for the establishment of three categories. Why not 5 categories? Why not different sleep durations as grouping factors? Authors could use the NSF recommendation for sleep duration as criteria, however, it would be even better if they presented the distribution of sleep duration in this population and based their categorization according to the frequency distribution in this population.

Response#5:

Thanks for your accurate comments about the issue. This is actually one of the weak points of our study, and most other similar researches were also not mentioned about this issue.

The proper sleep duration is not clearly defined in different ages. National Sleep Foundation (NSF) stated proper sleep duration in adults (26-64 years old) is 7-9 hours; and older adults (65+ years old) is 7-8 hours. On the other hand, American Academy of Sleep Medicine (AASM) also considered sleep duration 7 or more hours is proper for adult(>18 years old).

However, the average sleep duration in the community is about 6-7 hours (the residents usually get up really early such as 3:00-5:00am). Great majority of the residents in this community sleep about 6 hours per night. Because of the limited sample size, we only classified the population into three groups (very few people in the group of >7 hours and < 6 hours in the community) and set 6-7 hours in the middle group. This concern can be treated as one of limitations in our study.

Comment#6

The categorization for serum lipid profiles also could be better explained. Considering the NCEP ATP III guidelines there are several categories for each lipid (low, normal, borderline, high or very high). Authors decided by a binary categorization having just hyperlipidemia or no hyperlipidemia as categories. I'm a bit concern about this categorization and the lack of information in this case. Authors should take advantage of the sample-size and explore in more detail the lipid profile of their population. Furthermore, authors could perform analysis considering the data as a continuous variable and not as a grouping factor.

Response#6

Thanks for reviewer's comments and suggestions. In response to the reviewer's suggestions, the reason we chose binary categorization for no hyperlipidemia group (low and normal) and hyperlipidemia group (borderline, high or very high). Thanks for reviewer for pointing that we should explore more detail lipid profile of the population.

Comment#7

Regarding the questionnaires, how is it? How was evaluated the alcohol intake? Same for smoking and exercise. It is not clear how these factors were accessed. Furthermore, was there any evaluation of cardiometabolic disturbances? If so, authors must make it clear at manuscript.

Response#7

We thank for the reviewer for allowing us to making further explanation about it.

Our questionnaire contained the variable (presented as frequency in characteristic table) of "alcohol drinking" only defined as drinking times were greater or equal to 2 days per week.

Regular exercise in our questionnaire was defined as exercise times were greater or equal to 2 days per week.

Current smoking was defined as literal meaning.

We clarify the definitions in the Page 6 line 56-57.

Comment#8

Procedures: there is no clear description about the timeline of the study. Were the evaluations (physical, blood sampling, and questionnaires) performed at same day for every subject? If not, please describe it.

Response#8

We thank for reviewer's comments and suggestions. During the research, all participants were performed in the same day recorded from March 2014 to August 2014. We add it in Page6 line 41-42.

Comment#9

Statistical analysis and results:

The main statistical analysis are ok for the aim of the study. However, I have a few concerns regarding the strategy authors have use Spearman correlation to evaluate association between the serum lipid profile and several variables such as sex, current smoking and exercise. These variables are (or seem to be) binary. How could Spearman's correlation being performed for it? Still in regard of the correlation analysis: if the association between sleep duration and HDL is a U-shaped curve, how can authors explain the negative correlation presented at Table 3?

Regarding the Figure 1 the X axis labels are different from the statistical analysis. Please verify.

I will make no further comments on Discussion and conclusion considering my concerns on methods.

Response#9

We thank the reviewer for the comments and the reviewer's concern is well taken. We agree with the reviewer's opinion. It is not appropriate to use Spearman rank correlation to evaluate association between the serum lipid profile and binary variables (should use unpaired t-test instead). We acknowledged that our correlation analysis (Table 3) had several major weaknesses, especially it is an unadjusted analysis. We cannot announce any finding with the adjustment of possible confounding variables. Therefore, the authors decided to remove Table 3 which may confuse the readers.

Reviewer 2

Comment#1

Reference '10' has been cited for this sentence. This citation seems more suited to the next sentence and should be placed on line 14.

Response#1

We thank the reviewer for the correction about better citation. It is modified according to your suggestion.

Comment#2

'400' participants were enrolled however the numbers in the next line add up to 391 participants (141 men and 250 women). Please account for the apparent discrepancy.

Response#2

Thanks for reminding us about the typo. The correct enrolled numbers were 141 men and 259 women. It was corrected in the Page5 line 27.

Comment#3

Were there any specific inclusion/exclusion criteria for participants? Were patients with known sleep related disorders also included (for example obstructive sleep apnoea).

Van den Berg et al. (reference 16) in the discussion section of their paper did suggest that sleep apnoea did influence their results (albeit in a minor way) however there is ample literature available detailing the effects of sleep apnoea on cardiometabolic health. It is important to denote if patients with this condition were included or not.

Response#3

Thanks for reminding us about no clearly statement about inclusion/exclusion criteria as another reviewer's comment. The inclusion criteria included (1) the residents aged over 50 years old; (2) the residents living in Guishan township. 12 people aged younger than 50 years old were excluded from the study.

Subjects were excluded if (1) the residents could not complete the full examinations or missing data for age, sex, anthropometric values and blood test results; (2) the residents were functionally dependent; (3) the residents were unable to adequately communicate with the interviewers; (4) the residents had major illness recently; (5) the residents with known sleep disorders.

We have added a paragraph to the "MATERIALS AND METHODS" section in in Page 5 line 27-33.

Comment#4

Line 18 and 19:

Blood sampling was done to measure for lipid profile, AST levels, fasting plasma glucose, creatinine and uric acid levels. In this population, using the numbers given in the table, the prevalence of diabetes was 20%. Glycatedhaemoglobin would have been a more appropriate measurement of glycaemia especially given the already high prevalence of diagnosed diabetes in the group. Are HbA1c results available to report?

Response#4

In response to the reviewer's problem, we are sorry for that the research design including no measurement glycosylated hemoglobin. It is actually a proper way to define diabetes in a community; it should be acknowledged as a limitation and we list it on the "DISCUSSION" section in Page 20 line 198-200.

Comment#5

Line 21:

'Other lifestyle habits': Were there any questions relating to dietary habits? I notice in this study as well as other similar studies in the literature covering examining effects of sleep duration on serum lipids, diet does not seem to be factored for as a covariate. Whereas I appreciate the difficulty in trying to ascertain dietary habits in a study such as this one, was there any information at all available? Perhaps this could be acknowledged and expanded upon within the discussion.

Response#5

Thanks for your accurate comment about the issue. Actually our questionnaire including some diet habits such as vegetarian or not, the frequency of eating vegetables and fruits, three meals a day at regular hours and in a fixed quantity or not. However, the questions were difficult merging into one question and bringing to the regression model for adjustment. It should be acknowledged as another limitation and discussed on the "DISCUSSION" section in Page 20 line 200-204.

Comment#6

Line 26:

The definition of hyperlipidaemia is acceptable but in terms of lipid lowering therapy, what adjustment was made for these during analysis of the data? This is very pertinent given that 65% of participants in this study had a diagnosis of dyslipidaemia so I would assume a fair proportion of the participants were on lipid-lowering therapy. Commonly used drugs such as statins and fibrates can both influence HDL-C which was the main finding reported but additionally use of these drugs would have influenced the results on other lipid profile parameters. The use of lipid lowering therapy should be accounted for when discriminating findings between groups and adjusted for in the regression model.

Response#6

We thank for the reviewer for allowing us to making further explanation about it. Hyperlipidemia was defined as having LDL-C, TC, or TG levels above the cutoff point, but we did not considering about lipid-lowering medication while formulating the questionnaire. This concern can be treated as one of limitations in our study and discussed on the "DISCUSSION" section in Page 20 line 204-206.

Comment#7

Line 34:

If possible, could units of alcohol consumed per week be reported as opposed to simply days per week. Though drinking 2 or more days per week is likely to be a very inclusive parameter, there could be binge drinkers that are omitted by excluding units. If alcohol consumption (by units) information is not available, the current definition will suffice.

Response#7

In response to the reviewer's suggestion, our questionnaire contained the question about the kind of alcohol drinking (beers, wine, hard liquors such as Sorghum, medical wine...), how many times of alcohol drinking per week, and the amount of alcohol drinking per day (milliliter). However, the variable of "alcohol drinking" only defined as drinking times were greater or equal to 2 days for simplifying the question.

It maybe proper if calculated the alcohol equivalent for each participants instead of current question, but many alcohol drinkers took different drinking and forget how may milliliters they took per day, so it difficult to measure the alcohol equivalent. Therefore, we only set drinking times as our variable for brining to the regression model. We sincerely hope that the reviewers could approve our explanation.

Comment#8

Line 35:

Likewise, with exercise, a more precise definition would be getting an estimate of hours per week. Improved precision of the definitions may improve the accuracy of the regression models.

Response#8

The reviewer is correct regarding this problem. Regular exercise in our questionnaire was indicated as exercise times were greater or equal to 2 days per week. We clarify the definitions in the Page6 line 56.

Comment#9

Table 1:

The waist to hip ratio reported between the dyslipidaemic and non-dyslipidaemic group are exactly the same however the p-value is 0.003. Please double check this (whereas I do appreciate that you may have reported it this way to reduce the decimal points, the result does seem odd in its current form).

Response#9

Thanks for reminding us about the carelessness. The original data were 0.52 ± 0.05 (hyperlipidemia) and 0.54 ± 0.06 (non- hyperlipidemia), the p-value was 0.003; we simplified the data to one decimal place by standard rounding, so the 0.52 and 0.54 all rounded down to 0.5, and 0.05 and 0.06 rounded up to 0.1. Sorry for the negligence, and we corrected it in Table 1.

Comment#10

Line 66:

Please give details of the chosen post-hoc tests (e.g. Bonferroni etc.) in the methods section. In addition, for this point only, please specify where the differences were significant in post-hoc testing, that is, between which groups.

Response#10

The comment was absolutely right and thanks for the reviewer pointed the important issue. We chose Bonferroni correction for post-hoc analyzing the low HDL-C. There were significant difference between <6 hours group and >7 hours group (p value = 0.009), 6-7 hours and >7 hours (p value=0.001); no significant difference between <6 hours group and 6-7 hours (p value =1.0). We add the complementary in the Page 10 line 87-88.

Comment#11

Line 75:

Please include number of participants in each group in this section, i.e., how many were older than 65?

Response#11

In response to the reviewer's question, there were 168 people over 65 years old in the total participants. We add the complementary in the Page 13 line 99-100.

Comment#12

Line 85 and table 3:

Which sex?

Response#12

The coding of sex was 1 for male and 0 for female.

Comment#13

Table 3:

It is interesting that the factors that affected HDL-C (apart from sleep duration) also affected triglycerides in an inverse manner. Sleep duration oddly had a negative correlation coefficient for both serum HDL-c and triglycerides (though non-statistically significant) which is not in keeping with the direction of association for the other covariates.

Similarly, in the regression models, though the model for serum triglycerides had a non-significant p-value, the trend was for a reduced risk of high triglycerides in a U-shaped manner whereas for a low-HDL-c, this U-shaped distribution represented an increased risk. This is not in keeping with the known relationship between HDL-C and triglycerides which is inverse and the expected result would have been a similar distribution in this U-shaped curve for a given factor (as in table 3 for variables other than sleep duration). Interestingly, as the authors later cite in the discussion, a study by Choi et al. (reference 11) also alludes to this unexpected relationship when sleep duration is examined as a variable (especially in the over 60 age group). These findings are not supported by the study by Bjorvatn (reference 13) in which the U-shape curves follow a similar direction (for incidence of high triglycerides and low HDL-c according to sleep duration) albeit with less pronounced odds ratios with longer duration of sleep for triglycerides. Whereas in both this study and the study by Choi et al., the difference in triglycerides between groups is not statistically significant, there is a clear tendency for this association and I would welcome a slightly more expanded discussion around this.

Response#13

Thanks for your comments. We acknowledged that our correlation analysis (Table 3) had several major weaknesses, especially it is an unadjusted analysis. We cannot announce any finding with the adjustment of possible confounding variables. Therefore, the authors decided to remove Table 3 which may confuse the readers.

Comment#14

Line 134:

One of the references chosen (reference 18) refers to a study done in a paediatric population. Whereas it does support the author statement, a more suitable reference would be from adult medicine for which there are many.

Response#14

Thanks for the reviewer's suggestion. We delete the original reference 18 and add new one in Page 18 line 159(reference 25).

Comment#15

Line 137:

Please cite references for statements relating to the effects of smoking, alcohol and exercise on the lipid profile.

Response#15

Thanks for the reviewer's suggestion. We add new reference about smoking, alcohol to lipid profiles in Page 18 line 162(reference 26-28), and exercise to lipid profiles in Page 18 line 162 (reference 29-30).

Comment#16 References: Reference 5 – 'the' should read 'The'. Reference 11 – Please delete the brackets after International Journal of Obesity: '(2005)'.

Response#16 Thanks for the reviewer's comment. We correct it as your suggestion.